# PD-L1–PD-1 Pathway in the Pathophysiology of Multiple Myeloma

**DOI:** 10.3390/cancers12040924

**Published:** 2020-04-10

**Authors:** Hideto Tamura, Mariko Ishibashi, Mika Sunakawa-Kii, Koiti Inokuchi

**Affiliations:** 1Division of Diabetes, Endocrinology and Hematology, Department of Internal Medicine, Dokkyo Medical University Saitama Medical Center, Saitama 343-8555, Japan; 2Department of Hematology, Nippon Medical School, Tokyo 113-8603, Japan; m-sunakawa@nms.ac.jp (M.S.-K.); inokuchi@nms.ac.jp (K.I.); 3Department of Microbiology and Immunology, Nippon Medical School, Tokyo 113-8603, Japan; mariko-ishibashi@nms.ac.jp

**Keywords:** PD-L1 (B7-H1), PD-1, multiple myeloma, monoclonal gammopathy of undetermined significance (MGUS), immune checkpoint inhibitors, AKT pathway

## Abstract

PD-L1 expressed on tumor cells contributes to disease progression with evasion from tumor immunity. Plasma cells from multiple myeloma (MM) patients expressed higher levels of PD-L1 compared with healthy volunteers and monoclonal gammopathy of undetermined significance (MGUS) patients, and its expression is significantly upregulated in relapsed/refractory patients. Furthermore, high PD-L1 expression is induced by the myeloma microenvironment and PD-L1^+^ patients with MGUS and asymptomatic MM tend to show disease progression. PD-L1 expression on myeloma cells was associated with more proliferative potential and resistance to antimyeloma agents because of activation of the Akt pathway through PD-1-bound PD-L1 in MM cells. Those data suggest that PD-L1 plays a crucial role in the disease progression of MM.

## 1. Introduction

Multiple myeloma (MM) remains an incurable disease even with new treatment strategies using proteasome inhibitors, immunomodulatory drugs (IMiDs) and monoclonal antibodies. Ten percent of monoclonal gammopathy of undetermined significance (MGUS) patients develop MM and related disorders at 10 years [1]. MM patients with a prior diagnosis of MGUS had a prognosis similar to those without known MGUS [2]. The survival time is a median 2–3 and 5–7 years in MM patients with high-risk cyotogenetic abnormalities, i.e., IgH translocations (t(4;14), t(14;16)) and 17p13 deletion, and standard-risk MM patients, respectively [3]. It is important to elucidate the mechanism of disease progression from MGUS to MM, and new treatment strategies including immunotherapy are needed to inhibit progression and improve prognosis in MM patients [4,5].

PD-L1, which we first identified as a homologue of B7 family molecules [6,7], is expressed on immune cells, i.e., dendritic cells (DCs) and B and T cells, and controls T-cell immune responses such as peripheral tolerance, termination of immune responses and immune exhaustion after prolonged exposure to an antigen stimulus [8,9,10]. B7-1 (CD80) and PD-1 were identified as receptors of PD-L1, and PD-1, a 55-KDa transmembrane protein belonging to the CD28/CTLA-4 family, has an immunoreceptor tyrosine-based inhibitory motif in its intracellular domain. PD-L1−PD-1 ligation activates PD-1 downstream from SHP-2 and dephosphorylates ZAP70, which interferes with T-cell receptor signaling, leading to the inhibition of T-cell activation [11,12,13,14,15]. PD-L2, the other ligand for PD-1, is expressed in more restricted cells, such as DCs and macrophages, after activation [10,16]. In tumor immunity, most tumor cells express PD-L1 on their surface, and PD-L1 can bind to PD-1 expressed on cytotoxic T lymphocytes (CTLs), resulting in the escape from immune surveillance by the induction of CTL apoptosis [17,18,19].

Consistent with immune evasion by PD-L1, patients with PD-L1 expression on tumor cells had shorter survival times, compared with other patients, in many types of cancer [20,21,22]. In hematologic malignancies, PD-L1^+^ patients with diffuse large B cell lymphoma had shorter overall survival [23]. Immune checkpoint inhibitors, such as anti-PD-1 and anti-PD-L1 antibodies, recovered tumor immune surveillance by tumor-specific CTLs and achieved 12–28% overall response rates, resulting in the improvement of survival in refractory cancer patients with melanoma, renal cell carcinoma and non-small cell lung cancer [24,25]. These results demonstrate the emergence of a promising new immunotherapeutic approach for treatment-refractory cancer patients. In hematologic malignancies, the anti-PD-1 antibody nivolumab induced an approximately 66–87% overall response in relapsed/refractory patients with Hodgkin lymphoma, in which alterations in chromosome 9p24.1 increase the level of PD-1 [26,27].

This review describes the expression and function of the PD-1–PD-L1 pathway in MGUS and MM, and the possibility of immunotherapy through the blockade of this pathway.

## 2. Expression of PD-L1 in MGUS and MM

Leukemia and lymphoma cells express various levels of PD-L1 on their surface [28,29,30,31,32,33,34]. We analyzed PD-L1 expression in 14 human MM cell lines (HMCLs) and detected PD-L1 mRNA expression in nine using RT-PCR, although only four cell lines (KMS-12BM, KMS-34, RPMI8226 and U266) expressed cell-surface PD-L1 in flow cytometry [35]. In primary cells from patients, a high expression of PD-L1 was detected in 25% of 40 samples from newly diagnosed MM patients, although its expression on plasma cells from healthy volunteers and MGUS patients was lower [35,36,37]. PD-L1 expression on malignant plasma cells was also associated with an increased risk of disease progression from smoldering to symptomatic MM [37]. Furthermore, the expression levels were increased in relapsed/refractory MM patients compared with newly diagnosed ones [35,38], and samples from those with minimal residual disease (MRD) expressed higher levels of PD-L1 compared with those at diagnosis [38], suggesting that PD-L1 expression on MM cells may be involved in disease progression. However, other groups reported no difference in PD-L1 expression levels between MM and MGUS patients and healthy volunteers [38]. Kelly et al. found that the PD-L1 transcript levels in MM patients were equivalent to those in normal plasma cells [39], suggesting that PD-L1 protein expression may be post-transcriptionally regulated.

In the bone marrow microenvironment of myeloma, other cells expressed PD-L1. Myeloid and plasmacytoid DCs expressed significantly higher levels of PD-L1 compared with those in peripheral blood from healthy controls, and there was a significant correlation in the percentage of PD-L1^+^ cells between MM cells and CD141^+^ myeloid DCs [40]. In addition, some relapsed MM patients had high PD-L1 expression on myeloid-derived suppressor cells (MDSCs) [41], although PD-L1 expression was not detected on CD138^+^ cells in normal bone marrow [42]. 

The expression of PD-1, a receptor of PD-L1, was increased on T and natural killer (NK) cells in MM patients compared with its expression in healthy volunteers [43,44]. In patients with MRD, T cells expressed higher levels of PD-1 compared with those at diagnosis [38]. In a murine myeloma model using 5T33 cells, in which CD138^+^PD-L1^+^ myeloma cells were found in the bone marrow and spleen, the percentages of PD-1^+^ cells in CD8^+^ and CD4^+^ T cells in the spleen were 20–60% and 18–40% in the advanced stage, respectively, although those percentages were only 5–10% in control mice. Furthermore, the number of PD-1^+^CD8^+^ T cells was increased in the bone marrow and spleen, although PD-1 expression was very low on T cells in the peripheral blood and lymph nodes, indicating that MM cells might induce PD-1 expression on tumor-infiltrating lymphocytes. In those myeloma mice, PD-1^+^CD8^+^ T cells were susceptible to the induction of apoptosis and had decreased production of IL-1, TNF-α and IFN-γ. In addition, the proportion of PD-1^+^ NK cells, regulatory T cells and MDSCs was significantly increased compared with that in controls [42]. In bone marrow obtained from MM patients, the percentage of PD-1^+^ cells in CD8^+^ T cells was significantly increased compared with that from MGUS/smoldering MM patients [45]. In addition, our results showed the negligible expression of PD-L2, another ligand of PD-1 (in submission). 

## 3. Induction of PD-L1 Expression in MGUS and MM

Various mechanisms of PD-L1 induction on cancer cells were reported to be associated with proinflammatory cytokines and oncogenic and transcriptional pathways, such as PTEN, mTOR or PI3K [8,46,47,48,49]. PD-L1 protein expression was restricted, while PD-L1 mRNA expression was detected in most normal tissues, suggesting the mechanism of the post-transcriptional regulation of PD-L1, i.e., deacetylation, palmitoylation, microRNAs (miRNAs) and phosphorylation-dependent proteasome degradation induced by glycogen synthase kinase 3β (GSK3β) (Table 1) [19,46,49,50,51,52,53,54,55,56]. miRNAs such as mi-RNA34a, miRNA-200, miRNA-513 and miRNA-570 decrease PD-L1 expression on tumor cells, which confers T-cell immunity in the tumor microenvironment [46,57]. Not only epidermal growth factor (EGF)-mediated glycosylation, which could inactivate GSK3β, but also COP9 signalosome 5 (CSN5)-mediated deubiquitination induced by TNF-α, stabilized PD-L1 expression on tumor cells [58,59,60]. Ogawa’s group reported that *PD-L1 3’*-untranslated region disruption could elevate PD-L1 expression in multiple cancers [61].

In MM, several regulatory pathways were shown to be involved in PD-L1 upregulation on myeloma cells (Figure 1). For example, its upregulation was induced by IFN-γ or TLR-stimulated STAT1 activation through the MyD88/TRAF6 or MEK/ERK pathway [36]. IFN-γ produced by several types of immune cells, i.e., activated Th1 (CD4^+^ helper T cells), CD8^+^ CTLs, macrophages, γδ T cells, NK cells and NKT cells, activated the JAK/STAT pathway, resulting in the induction of PD-L1 via IRF1 expression, although the amount of intrinsic IRF1 was not sufficient to induce PD-L1 expression [62,63,64,65,66]. In addition, MM cells expressed high levels of TLR2, TLR4, TRL7 and TLR9 [67,68]. We found that bone marrow stroma cells (BMSCs) also induced PD-L1 expression on MM cells by BMSC-derived soluble factor IL-6 through the IL-6 signaling cascade, including JAK2, STAT3 and the MEK1/2 pathway [35]. Consistent with these results, the JAK inhibitor ruxolitinib downregulated PD-L1 expression on MM cells [69,70]. It was reported that a proliferation-inducing ligand (APRIL), secreted by eosinophils, osteoclasts and myeloid cells in the MM microenvironment, could bind B-cell maturation antigens (BCMAs) expressed on MM cells, resulting in tumor cell survival as well as the upregulation of PD-L1 expression via the phosphorylation of MEK1/2 [71,72]. Those findings demonstrated that PD-L1 expression could be induced in the MM microenvironment.

The upregulation of PD-L1 expression on MM cells is induced by interleukin (IL)-6 and interferon (IFN)-γ, which are produced by the MM microenvironment. PD-L1 delivers PI3K/Akt signals to MM cells through PD-1 binding, contributing to the proliferative potential and drug resistance against antimyeloma agents. The soluble form of PD-L1 may contribute to the escape from immune evasion. APRIL, induced by eosinophils, osteoclasts and myeloid cells in the MM microenvironment, binds B-cell maturation antigens (BCMAs) on MM cells, resulting in tumor cell survival as well as the upregulation of PD-L1 expression via the phosphorylation of MEK1/2.

During the clinical course of MM, PD-L1 expression on monocyte, myeloid and plasmacytoid DCs was increased during treatment without daratumumab, although daratumumab treatment prevented PD-L1 upregulation in patients with MM [73]. Our in vitro study showed that lenalidomide and pomalidomide, but not thalidomide, induced PD-L1 expression on IMiD-insensitive MM cell lines and plasma cells from relapsed/refractory MM patients through the BCMA–APRIL pathway. We found that IMiDs induced APRIL expression through Ikaros degradation mediated by cereblon in MM cells [74]. Histone deacetylase inhibitors (HDACis) were also reported to induce PD-L1 expression on MM cells, indicating that HDACis combined with the anti-PD-L1 antibody enhance anti-MM immunity [75]. 

## 4. Functions of PD-L1 in MM; a Disease Progression Enabler?

In general oncology, PD-L1 is expressed on most tumor cells and binds to PD-1 on tumor-specific CTLs, resulting in the recruitment of SHP-1 and SHP-2 to the C-terminal of the PD-1 intracellular domain, leading to the inhibition of downstream PI3K/Akt signaling, which downregulates cell survival gene Bcl-xL expression and induces the apoptosis of CTLs. Thus, the PD-L1−PD-1 signaling pathway has a crucial role in tumor immune escape. 

Numerous pathogenetic and potential pathways involved in the pathophysiology of MM have been reported [76]. The interactions between MM cells and bone marrow stromal cells are associated with the pathophysiology of MM. This interaction in the MM microenvironment induces the secretion of several cytokines, such as IL-6, IGF-1, VEGF, SDF1α, TGF-β, HGF and TNF-α, derived from both BMSCs and MM cells, leading to the activation of multiple signaling pathways in MM cells. These include important pathways, such as PI3K/Akt, JAK/STAT3, RAS/RAF/MEK/ERK and NF-κB, for cell growth, anti-apoptosis and drug resistance in MM cells.

As in other tumors, PD-L1-expressing myeloma cells can inhibit the activity of CTLs, resulting in immune evasion. In addition, plasmacytoid DCs with high expressions of PD-L1 inhibit the activation of T and NK cells [77]. In a myeloma model, BMSCs promoted MM cells, likely by suppressing CD4^+^ T cells via the PD-L1−PD-1 pathway, indicating that the anti-PD-1 antibody may inhibit BMSC-induced MM cell growth [78]. We found that PD-L1-expressing myeloma cells exhibit aggressive behavior, with not only greater proliferative potential, but also more resistance to antimyeloma drugs [35]. PD-L1^+^ fractions in RPMI8226 and KMS-34 MM cells had more proliferative cells in the G2/M phase and high BrdU incorporation as well as Ki-67 and BCL2 expression and more rapid proliferation in culture compared with PD-L1^−^ fractions [35]. PD-L1 knockdown in highly PD-L1-expressing MM cell lines resulted in a significant inhibition of tumor cell proliferation and an increase in melphalan-induced apoptosis. We also found that PD-L1 binding with PD-1-Fc beads in myeloma cells induced resistance to anti-myeloma agents, such as melphalan and bortezomib, accompanied by the activation of the PI3K/AKT signaling pathway, because PD-1−PD-L1 bingeing delivered a reverse signal from PD-L1 into myeloma cells with the upregulation of CCND3, BCL2 and MCL1 [79]. PD-L1^+^ cells had higher Ki-67 and BCL2 expression levels, and highly positive PD-L1 cells were associated with high levels of serum LDH and the percentage of plasma cells in the bone marrow of MM patients, indicating that PD-L1 may be associated with MM disease progression [35]. Consistent with those results, in MGUS and asymptomatic MM, a higher expression of PD-L1 on MM cells was correlated with an increased risk of progression to clinical MM [37]. The expression levels of PD-L1 were also increased during smoldering myeloma progression [80].

## 5. Soluble Form of PD-L1 in MM Patients

The soluble form of PD-L1, which is released from surface PD-L1 on tumor cells into the circulation, is also thought to exert immunosuppressive activity [81]. Frigola et al. reported that soluble PD-L1 from tumor cell lines had a molecular weight of 45 kDa and an Ig-V ligand-binding domain that could bind to PD-1 [78]. High soluble levels of serum PD-L1 were associated with poor prognosis in patients with renal cell cancer, hepatocellular carcinoma, gastric cancer and B cell lymphoma [81,82,83,84]. In MM patients, high serum soluble PD-L1 levels were reported to be associated with advanced R-ISS stage and shorter progression-free and overall survival times [85,86]. High soluble PD-L1 levels in bone marrow plasma from MM patients at 100 days after autologous hematopoietic stem cell transplantation was an independent risk factor for a short response period [87]. Some tumor cells, such as melanoma and lung and breast cancer cells, have been known to release extracellular vesicles, mostly in the form of exosomes, which could include PD-L1 [88]. Exosomal PD-L1 also has immune-suppressive functions through the suppression of T-cell immunity. In vivo experiments showed that exosomal PD-L1-derived tumor cells promoted the growth of PD-L1-knockdown tumors. Thus, myeloma-derived extracellular vesicles might inhibit anti-tumor immune responses by PD-L1 delivery, as well as the expansion of regulatory T cells, stimulation of M2 macrophage polarization and induction of tolerogenic dendritic cells [89]. These results suggest that soluble PD-L1 may be associated with disease progression by impairing host immunity.

## 6. Blockade of the PD-L1−PD-1 Pathway as a Therapeutic Target in MM 

Considering that the PD-L1−PD-1 pathway may be associated with the pathophysiology of MM, anti-PD-1/PD-L1 antibody treatment could be clinically effective in MM patients by recovering T-cell cytotoxicity and inhibiting reverse signaling from PD-L1 on MM cells. Furthermore, DCs also expressed PD-L1 in the MM microenvironment and then suppressed PD-1-expressing T-cell and NK-cell immune functions via the PD-L1/PD-1 pathway [40]. PD-L1 expression on plasmacytoid DCs, which play important roles in the cell growth and prolonged survival in MM cells, was increased and localized with PD-L1-expressing MM cells in bone marrow [90]. In addition, PD-L1 expression on monocytes obtained from MM patients was increased compared with those from healthy volunteers [70]. PD-L1 on those cells as well as MM cells may be a therapeutic target of the PD-L1−PD-1 blockade. 

In vivo experiments using a murine myeloma model demonstrated that both the PD-1 deficiency and PD-L1 blockade inhibited tumor growth [43]. Furthermore, the anti-PD-L1 antibody enhanced T-cell immune responses when used in combination with other therapies, such as DC-myeloma fusion vaccine and irradiation [44,91]. In the 5T33 murine MM model, autologous stem cell transplantation in combination with vaccination could prolong survival in those mice when the PD-L1 blockade was added [42]. An in vivo murine myeloma model showed that low-dose irradiation followed by anti-PD-1 treatment prolonged survival with increased tumor-specific CD8^+^ T cells, in which both CD4^+^ and CD8^+^ T cells were necessary to eliminate MM cells [91]. These reports suggested that the blockade of the PD-1/PD-L1 pathway, combined with the induction of tumor-specific CTLs by other immunotherapy, radiation or chemotherapy, may have great promise as a treatment for MM patients [4,92].

The results of a nivolumab phase 1B trial showed that nivolumab had little effect, except for one complete response along with stable disease among 63% of the 27 MM patients enrolled [93]. This low effectiveness of the immune checkpoint inhibitor may be due to immune dysfunction with increased immunosuppressive cells such as regulatory T cells and MDSCs and exhausted T cells [94,95]. In MM patients with advanced-stage disease, T-cell signaling (CD28, CD152, CD3zeta, p56lck, ZAP-70, PI3-k) was impaired and its intracellular cytokine expression (IL-γ, IL-2, IL-4) was downregulated in CD4^+^ and CD8^+^ T cells [96], and PD-1^+^CD8^+^ T cells from the bone marrow of MM patients co-expressed other immune checkpoint inhibitory receptors such as RAG3 and TIGIT. Therefore, the anti-PD antibody alone did not result in the recovery of T-cell proliferation, indicating that the blockade of the PD-L1−PD-1 pathway does not markedly reverse T-cell exhaustion [45]. In a murine myeloma model, anti-TIGIT was more effective in improving survival compared with anti-PD-1 [97]. Anti-PD-1 combined with a TGF-β inhibitor could reactivate T cells from the bone marrow of MM patients, suggesting that the combination may be effective in treating refractory disease [45].

In addition, numerous types of T-cell dysfunction have been reported, i.e., a decreased number of CD4^+^ T cells, an abnormal ratio of CD4^+^/CD8^+^ T cells and an imbalance of Th1/Th2 [98,99,100]. Suen et al. showed that the clonal expansion of CD8^+^ TCRVβ^+^CD57^+^CD28^–^ T cells, which are thought to be tumor specific, had a lower PD-1 expression compared with CD8^+^TCRVβ^–^CD57^+^ nonclonal T cells, indicating low binding of the anti-PD-1 antibody [101]. Furthermore, immune checkpoint inhibitors are effective in patients with cancers exhibiting frequent nonsynonymous mutations, which could induce tumor-specific CTLs, although myeloma has a lower frequency of nonsynonymous mutations than cancers that have been approved as treatment targets of immune checkpoint inhibitors [101].

IMiDs have various immunomodulatory effects, i.e., the activation of T and NK cells, inhibition of regulatory T cells and MDSCs, increased IL-2 production and Th1 cells, and the fact that lenalidomide decreases PD-1 expression on T cells [43,102,103,104]. Thus, IMiDs could be reasonable in combination with immune checkpoint inhibitors for treating MM patients. In vitro experimental results showed that the anti-PD-1/PD-L1 antibody combined with lenalidomide could induce the apoptosis of MM cells and increase IFN-γ production in CD4^+^ and CD8^+^ T cells [41]. It was also reported that the anti-PD-1 antibody increased chemotaxis through the SDF-1/CXCR4 axis, resulting in the enhancement of NK cell-mediated cytotoxicity against MM cells [43,44], and those results support the reasonable combination of anti-PD-1 and IMiDs. Consistent with that concept, treatment with the anti-PD-1 antibody pembrolizumab combined with lenalidomide or pomalidomide achieved a 44% or 60% overall response rate in relapsed/refractory MM patients, respectively [105,106] (Table 2). However, patients treated with pembrolizumab and pomalidomide had a shorter progression-free survival time and a higher incidence of treatment-related deaths (3%), including neutropenic sepsis, myocarditis and Stevens–Johnson syndrome, resulting in the suspension of clinical trials of the anti-PD-1 and IMiD combination [107]. 

Clinical trials using other combinations of immune checkpoint inhibitors with immunotherapies are ongoing. A clinical trial of the anti-PD-1 antibody nivolumab combined with the anti-CD38 antibody daratumumab is underway but not recruiting (NCT01592370). A murine myeloma model using 5T33 cell lines showed that a whole-tumor cell vaccine combined with the anti-PD-L1 antibody increased the antimyeloma effects compared with tumor-cell vaccination alone or the anti-PD-L1 antibody alone [42]. A clinical trial of the anti-PD-1 antibody pidilizumab, combined with a myeloma-specific DC fusion vaccine, which induces tumor-specific CTLs, is in progress without recruiting (NCT01067287). Treatment strategies based on the blockade of the PD-L1−PD-L1 pathway are being developed to prevent disease progression in smoldering MM and stable disease after high-dose chemotherapy with autologous stem cell transplantation (Table 3). It appears reasonable to control the disease using the T-cell immune surveillance recovered by PD-1/PD-L1 inhibitors, although several clinical trials have been suspended due to safety concerns.

## 7. Conclusions

The myeloma environment induces PD-L1 expression on myeloma cells as well as immune cells in the bone marrow, resulting in the inhibition of T-cell immunity. Furthermore, PD-L1^+^ myeloma cells exhibit intrinsic aggressive behavior independent of the immune evasion mechanism, because PD-L1 can deliver a reverse signal to MM cells via activation of the PI3K/AKT signaling pathway. Thus, the PD-1–PD-L1 pathway plays a crucial role in the pathophysiology of MM, as well as in disease progression from MGUS to MM, providing new potential immunotherapy approaches targeting the PD-L1 protein. The combination of the anti-PD-1/PD-L1 antibody with an IMiD led to a high mortality rate, which might be associated with an excessive autoimmune reaction. Other combination treatments, such as the anti-PD-1 antibody and radiation therapy or tumor vaccination, etc., may improve the prognosis of refractory MM patients.

## Figures and Tables

**Figure 1 cancers-12-00924-f001:**
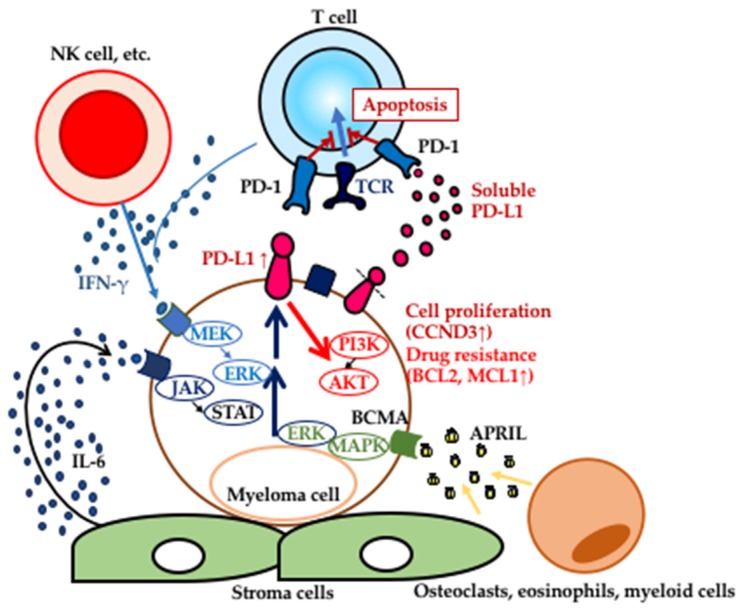
PD-L1 induced by the MM microenvironment.

**Table 1 cancers-12-00924-t001:** Mechanism of post-transcriptional regulation of PD-L1 on tumor cells.

**Downregulation of PD-L1**
Deacetylation
miRNAs (miRNA-34a, miRNA-200, miRNA-513, miRNA-570)
Phosphorylation-dependent proteasome degradation induced by GSK3β
**Stabilized PD-L1**
DHHC(Asp-His-His-Cys)3-dependent palmitoylation (inhibition of PD-L1 ubiquitination)
EGF-mediated glycosylation (GSK3β inactivation)
Reduction of PD-L1 degradation by EGF via B3NT3 upregulation
PTEN/PI2K/mTOR/S6K1 pathway
CSN5-mediated deubiquitination induced by TNF-α

miRNA, microRNA; GSK3β, glycogen synthase kinase 3β; EGF, epidermal growth factor; CSN5, COP9 signalosome 5.

**Table 2 cancers-12-00924-t002:** PD-1 inhibitors in combination with IMiDs in patients with relapsed/refractory MM.

Treatment	*N*	Prior Therapy Lines	Age, Median (Range), Years	Patient Characteristics	ORR/CR/VGPR	PFS/OS, Months	Adverse Events Grade ≥ 3
Nivolumab (Phase 1b) [85]	27	2–3; 44%4–5; 30%≥6; 22%	63(32–81)	-	4%/4%/0%	-	19%
PEM + Rd(Phase 1)KEYNOTE-023 [97]	62	0–1; 30.6%2–3; 35.5%≥4; 33.9%	61(46–77)	HRC 10%, LEN-refractory 76%, prior ASCT 87%	44%/4%/12%	7.2/N.R.	60% (neutropenia 27%, thrombocytopenia 16%, anemia 8%, hyperglycemia 7%)
PEM + Pd (Phase 2) [98]	48	Median 3, range 2–5	64(35–83)	HRC 62%, PI/IMiD-refractory 73%,prior ASCT 70%	60%/8%/27%	17.4/N.R.	40% (hematologic toxicities 40%, hyperglycemia 25%, pneumonia 15%)
PEM + Pd (Phase 3) KEYNOTE-183 [99]	125	≥2	65(45–94)	HRC 22%	34%	5·6 (8·4 in Pd) /N.R.	63% (treatment-related deaths 3%; unknown cause, neutropenic sepsis, myocarditis, Stevens–Johnson syndrome)

ORR, overall response rate; CR, complete response; VGPR, very good partial response; PFS, progression-free survival; OS, overall survival; PEM, pembrolizumab; Rd, lenalidomide + dexamethasone; LEN; lenalidomide; HRC, high-risk cytogenetics; ASCT, autologous stem cell transplantation; N.R., not reached; Pd, pomalidomide + dexamethasone; PI, proteasome inhibitor; IMiD, immunomodulatory drug.

**Table 3 cancers-12-00924-t003:** Clinical trials using blockade of the PD-L1−PD-1 pathway to prevent disease progression in multiple myeloma.

Treatment	Patients	Phase	Status	Identifier
**PD-1 inhibitors**				
Pembrolizumab	Intermediate or high-risk SMM	Early phase 1	Active, not recruiting	NCT02603887
Pembrolizumab	MM during lymphopenia post-HDT/ASCT	Phase 2	Completed	NCT02331368
Nivolumab + LEN + DEX	High-risk SMM	Phase 2	Suspended	NCT029003381
**PD-L1 inhibitors**				
Atezolizumab	High-risk asymptomatic MM	Phase 1	Suspended	NCT02784483
PD-L1 peptide vaccine	MM post-HDT/ASCT	Phase 1	Active, not recruiting	NCT03042793
PD-L1 peptide vaccine	High-risk SMM	Phase 2	Recruiting	NCT03850522

SMM, smoldering multiple myeloma; HDT/ASCT, high-dose chemotherapy with autologous stem cell transplantation; LEN, lenalidomide; DEX, dexamethasone.

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
