# Peer review of "PD-L1–PD-1 Pathway in the Pathophysiology of Multiple Myeloma"

_cancers, 2020, doi:10.3390/cancers12040924_

Round 1

Reviewer 1 Report

PD-L1–PD-1 pathway in the pathophysiology of multiple myeloma

The manuscript should be revised for minor linguistic and grammatical errors.

It is highly recommended to explain PD-L1–PD-1 pathway in general oncology. Then discuss its relation to pathophysiology of multiple myeloma.

It is recommended to discuss the different possible pathogenesis and potential pathways involved in pathophysiology of multiple myeloma.

The review cover some clinical trials related to the involvement of PD-L1–PD-1 pathway in the pathophysiology of multiple myeloma. It is highly recommended to include basic studies involving in vitro cell culture and in vivo animal studies related to the topic.

For sound review to be publishable in "Cancer", It is important to include more tables and figures summarizing the added parts during the review process.

Author Response

We thank the reviewers for their comments on our manuscript and appreciate their help in improving it. We have revised the manuscript in response to each comment and rewrote the highlights in the revised version. The revised text appears in red. We hope that the revisions are satisfactory and enable the manuscript to be accepted for publication in Cancers.

Reviewer 1:

1) The manuscript should be revised for minor linguistic and grammatical errors.

Our revised manuscript was edited by a professional English native-speaking specialist before resubmission.

2) It is highly recommended to explain PD-L1–PD-1 pathway in general oncology. Then discuss its relation to pathophysiology of multiple myeloma.

Thank you for your suggestion. We added the explanation of the PD-L1–PD-1 pathway in general oncology before describing its relation to the pathophysiology of MM (page 4, lines 145−149), although we have already noted this on pages 1–2, lines 41–46.

3) It is recommended to discuss the different possible pathogenesis and potential pathways involved in pathophysiology of multiple myeloma.

We added the different pathways involved in the pathophysiology in the text (page 4, lines 150–156) with an additional reference [76].

4) The review cover some clinical trials related to the involvement of PD-L1–PD-1 pathway in the pathophysiology of multiple myeloma. It is highly recommended to include basic studies involving in vitro cell culture and in vivo animal studies related to the topic.

We added data from basic studies on blocking the PD-L1–PD-1 pathway in myeloma (pages 5—6, lines 205—214) with an additional reference [92].

5) For sound review to be publishable in "Cancer", It is important to include more tables and figures summarizing the added parts during the review process.

We added Table 1 covering the added parts in the revised version.

Reviewer 2 Report

the minor issues:

In line 95, miss the reference.

In line 102, one of the post-translation modifications of PD-L1  is palmitoylation, which should be included.

In line 139, the title is not accurate, authors just mentioned human cancer cells,  there is not the relationship with patients

In line 160, authors discussed the sPD-L1, should include exosome of PD-L1

In line 172, before the discussion of blockade, PD-L1 expression in dendritic cells and macrophages of MM patients should be considered.

Author Response

We thank the reviewers for their comments on our manuscript and appreciate their help in improving it. We have revised the manuscript in response to each comment and rewrote the highlights in the revised version. The revised text appears in red. We hope that the revisions are satisfactory and enable the manuscript to be accepted for publication in Cancers.

Reviewer 2:

1) In line 95, miss the reference.

Those results are included in our submission data, and we cited it as "in submission" (page 3, line 94).

2) In line 102, one of the post-translation modifications of PD-L1 is palmitoylation, which should be included.

According to your valuable suggestion, we added "palmitoylation" in the mechanism of posttranscriptional regulation of PD-L1 (page 3, lines 100), which is also included in Table 1, and one reference [additional reference: 50].

3) In line 139, the title is not accurate, authors just mentioned human cancer cells, there is not the relationship with patients.

The title "Functions of PD-L1 in MM patients; a disease progression enabler?" was changed with the deletion of "patients" (page 4, line 144).

4) In line 160, authors discussed the sPD-L1, should include exosome of PD-L1.

Thank you for your great suggestion. We discussed PD-L1 delivery by extracellular vesicles including exosomes in the text [additional references: 88,89] (page 5, lines 187−193).

5) In line 172, before the discussion of blockade, PD-L1 expression in dendritic cells and macrophages of MM patients should be considered.

We noted the expression of PD-L1 on dendritic cells and monocytes in MM patients [additional refence: 90] (page 5, lines 198−204).

Reviewer 3 Report

In this review, Tamura et al. describe the role of PD-L1/PD-1 pathway in  the pathophysiology of multiple myeloma (MM) and the mechanisms that regulate PD-L1 overexpression on MM cells. PD-L1 expression correlates to MM cells proliferation and drug resistance. For these reasons, authors also introduce the importance of PD-L1 inhibition in MM.

The review is well-written, interesting and updated. I only suggest to include the importance of miRNA in the regulation of PD-1/PD-L1 expression (Wang et al., International Journal of Molecular Science. 2017; Tremblay-LeMay et al., Journal of Hematology & Oncology. 2018)

Author Response

We thank the reviewers for their comments on our manuscript and appreciate their help in improving it. We have revised the manuscript in response to each comment and rewrote the highlights in the revised version. The revised text appears in red. We hope that the revisions are satisfactory and enable the manuscript to be accepted for publication in Cancers.

Reviewer 3

1) I only suggest to include the importance of miRNA in the regulation of PD-1/PD-L1 expression (Wang et al., International Journal of Molecular Science. 2017; Tremblay-LeMay et al., Journal of Hematology & Oncology. 2018)

Thank you for your suggestion. We additionally mentioned the importance of miRNA in the regulation of PD-L1 expression and references [Wang's paper already cited; Tremblay-LeMay R, et al., J Hematol Oncol 2018 (#57)] (page 3, lines 101−103).

Round 2

Reviewer 1 Report

As the authors addressed the reviewers comments, I suggest acceptance of the manuscript.